



# Multipoint Observations of Compressional Pc5 Pulsations in the Dayside Magnetosphere and Corresponding Particle Signatures

Galina Korotova[1,2], David Sibeck[3], Mark Engebretson[4], Michael Balikhin[5], Scott Thaller[6], Craig Kletzing[7], Harlan Spence[8], and Robert Redmon[9]

[1]IPST, University of Maryland, College Park, MD, USA

[2]IZMIRAN, Russian Academy of Sciences, Moscow, Troitsk, Russia

[3]NASA/GSFC, Code 674, Greenbelt, MD, USA

[4] Department of Physics, Augsburg University, Minneapolis, MN, USA

[5]Department of Automatic Control and Systems Engineering, University of Sheffield, Sheffield, UK.

[6] LASP, University of Colorado, Boulder, CO, USA

[7]Department of Physics and Astronomy, Iowa University, Iowa City, IA, USA

[8]EOS, University of New Hampshire, Durham, NH, USA

[9]Solar and Terrestrial Physics division, NGDC/NOAA, Boulder, CO, USA

**Abstract**

We use Van Allen Probes Radiation Belt Storm Probes-A and -B (henceforth RBSP-A and -B) and GOES-13 and -15 (henceforth G-13 and G-15) multipoint magnetic field, electric field, plasma, and energetic particle observations to study the spatial, temporal, and spectral characteristics of compressional Pc5 pulsations observed during the recovery phase of a strong geomagnetic storm on January 1, 2016. From ~19:00 UT to 23:02 UT, successive magnetospheric compressions enhanced the peak-to-peak amplitudes of Pc5 waves with 4.5-6.0 mHz frequencies from 0-2 to 10-15 nT at both RBSP-A and -B, particularly in the prenoon magnetosphere. Poloidal Pc4 pulsations with frequencies of ~22-29 mHz were present in the radial Bx component. The frequencies of these Pc4 pulsations diminished with increasing radial distance, as expected for resonant Alfvén waves standing along field lines. The GOES spacecraft observed Pc5 pulsations with similar frequencies to those seen by the RBSP, but Pc4 pulsations with lower frequencies.

Both RBSP-A and -B observed frequency doubling in the compressional component of the magnetic field during the Pc5 waves, indicating a meridional sloshing of the equatorial node over a combined range in $Z_{SM}$ from 0.25 to -0.08 Re, suggesting that the amplitude of this





meridional oscillation was ~0.16 Re about an equatorial node whose mean position was near $Z_{SM}$
= ~0.08 Re.  RBSP-A and -B HOPE and MagEIS  observations provide the first evidence for a
corresponding frequency doubling in the plasma density and the flux of energetic electron,
respectively.  Energetic electron fluxes oscillated out of phase with the magnetic field strength
with no phase shift at any energy.  In the absence of any solar wind trigger or phase shift with
energy, we interpret the compressional Pc5 pulsations in terms of the mirror mode instability.
**Introduction**
ULF pulsations with periods of 100s or greater and high azimuthal wave numbers (m)
with magnetic field perturbations in the radial direction and electric field perturbations in the
azimuthal direction within the Earth's magnetosphere are typically poloidal waves [Sugiura and
Wilson, 1964].  According to Elkington et al. [2003], energetic particles with drift frequencies of
6.7-22 mHz and 1.7-6.7 mHz can readily interact with corresponding high-m poloidal Pc4 and
Pc5 pulsations.  Because the atmosphere and ionosphere screen these high-m waves from the
ground, they can only be studied with the help of satellite observations.  Thus studies like that of
Dai et al. [2013] employed observations from locations at or near geosynchronous orbit.  Higbie
et al. [1982] and Nagano and Araki [1983] showed that long-lasting compressional Pc5
pulsations occur most frequently in the dayside magnetosphere during the recovery phase of
magnetic storms. Storm-time Pc5 pulsations occur in the afternoon sector between 12:00 and
18:00 local time following injections of ring current particles [Kokubun, 1985].
A number of studies have examined compressional Pc5 waves outside geostationary
orbit.  According to these studies, compressional Pc5 waves were observed in the dawn
[Hedgecock, 1976], dusk [Constantinescu et al., 2009] and noon [Takahashi et al., 1985] sectors.
Zhu and Kivelson [1991] reported that intense compressional waves are a persistent feature on
both flanks of the magnetosphere. Compressional Pc5 pulsations occur within ~20° region of the
magnetic equator [Vaivads et al., 2001].  They have several Re wavelengths [Walker et al.,
1982] and often exhibit harmonics.  Elkington et al. [2003] noted that poloidal and
compressional modes are far more effective the radial transport of energetic particles than the
toroidal mode.  Two methods are used to identify the harmonic mode of a poloidal oscillation.
The first compares the phase difference between the radial component of the magnetic field and
the azimuthal component of the electric field [Takahashi et al., 2011].  The second compares
observed wave frequencies with the eigenfrequencies predicted by theory [Cummings, 1969].
The multi-satellite study of Takahashi et al. [1987a] showed that a compressional Pc 5 wave had
an antisymmetric standing structure.





Compressional Pc5 pulsations have been ascribed to numerous excitation mechanisms.
They can be produced by internal and external processes.  It is supposed that the solar wind is the
main external source for pulsations produced by the Kelvin-Helmholtz (KH) instability at the
magnetopause or the inner edge of the low-latitude boundary layer [e.g., Guo et al., 2010].
Observations indicating enhanced rates of Pc5 occurrence during periods of greater solar wind
velocity support this model [e.g., Engebretson et al., 1998].  Transient variations  in the dynamic
pressure of the solar wind or foreshock [e.g., Wang et al., 2018; Shen et al., 2018] that cause
abrupt changes in the magnetic field strength  in the magnetosphere and sudden impulses in the
ionosphere [e.g., Zhang et al., 2010, Sarris et al., 2010] provide another possible trigger for Pc5
pulsations.    External pressure impulses can cause compressional oscillations of the
magnetosphere with discrete eigenfrequencies, known as global modes or cavity/waveguide
modes [Samson et al., 1992].  Periodic solar wind dynamic pressure variations directly drive
some compressional   magnetospheric magnetic field oscillations [e. g., Kepko et al., 2003;
Motoba et al., 2003]. Takahashi and Ukhorskiy [2008] considered solar wind pressure variations
as the main external driver of Pc5 pulsations observed at geosynchronous orbit in the dayside
magnetosphere.
Internal generation mechanisms for compressional Pc5 pulsations include the drift-
bounce resonant instability which occurs for particles   with resonance drift and bounce periods
[Southwood et al., 1969] and the drift-mirror instability in the presence of strong temperature
anisotropies [Chen and Hasegawa, 1991].  In high β plasmas (β is the plasma pressure divided by
the magnetic pressure), these mechanisms favor antisymmetric waves [Cheng and Lin, 1987].
One of the possible mechanisms of generation of compressional Pc5 pulsations observed
at geosynchronous orbit is  a drift mirror instability of ring current particles [e.g., Lanzerotti et
al., 1969].  While the observed anticorrelated magnetic field strength and ion flux oscillations
are expected for a drift mirror wave [Kremser et al., 1981], the instability criterion is generally
not satisfied [Pokhotelov et al., 1986]. One possible reason for the  lack of consistency between
theory and observation might be because the real geometry of the magnetosphere is not taken
into account [Cheng and Lin, 1987].        Compressional pulsations are often accompanied by
pulsations in particle fluxes [Kremser et al., 1981; Liu at al., 2016].  Particle observations can
provide useful information on the spatial and wave structure of ULF pulsations.  Lin et al.
[1976] explained flux oscillations as the adiabatic motion of particles in a
magnetohydrodynamic wave.  Kivelson and Southwood [1985] studied charged particle
behavior in compressional ULF waves and showed that "a mirror effect" is the dominant cause





for particle flux modulations. Finite gyroradius effects enable detection of gradients in particle
flux   associated with waves [e.g., Korotova et al., 2013].

**1. Objectives**
We use multipoint magnetic field, plasma, and energetic particle observations from
RBSP-A and -B and G-13 and -15 to study the spatial, temporal, and spectral characteristics of
compressional Pc5 pulsations observed deep within the magnetosphere during the recovery
phase of the strong magnetic storm which began on December 31, 2015. We investigate the
mode of the waves and their nodal structure. We focus on the properties of double frequency
pulsations that occurred in the vicinity of the geomagnetic equator. We demonstrate that the
energetic particles respond directly to the compressional Pc5 pulsations and also exhibit a double
frequency oscillation.  We search for possible solar wind triggers and test two possible
generation mechanisms: drift-bounce resonance, and mirror instability. The paper is organized
as follows: Section 2 describes instruments and resources. Section 3 presents the solar wind and
IMF conditions. Section 4 provides an analysis of these waves and their generation mechanisms.
**2. Resources**
The Van Allen Probes mission can be used to study the geospace response to a
fluctuating solar wind. The  mission began in August 2012 with a twin spacecraft launch into
similar 10° inclination orbits with perigee altitudes slightly greater than 600 km and apogee
altitudes just beyond 30000 km [Mauk et al., 2012]. The spacecraft carry instruments that
measure electromagnetic fields, waves, and charged particle populations deep within the
magnetosphere. This paper employs observations of 20-4000 keV electrons from the MagEIS
instrument [Blake et al., 2013] in the Energetic Particle, Composition, and Thermal (ECT) suite
[Spence et al., 2013] in conjunction with observations from the magnetometer in the Electric and
Magnetic Field Instrument Suite and Integrated Science suite [Kletzing et al., 2013], and the
Electric Field and Waves (EFW) [Wygant et al., 2013] instrument. We examine electric and
magnetic field measurements with 11s and 4s time resolution, respectively, and differential
particle flux observattions with ~11s (spin period) time resolution. The data are provided by
NASA/GSFC's CDAWEB in the MGSE (modified GSE) coordinate system. We use magnetic
field data from G-13 and -15 with 0.5 s time resolution [Singer et al., 1996]. Finally, we employ
Wind solar wind magnetic field and 3DP plasma data with 3s time resolution [Lepping et al.,
1995; Lin et al., 1995].





### 3. Orbits and solar wind and geomagnetic conditions


The pulsation events to be studied here occurred late on January 1, 2016, following a
prolonged period of strongly southward IMF orientation and geomagnetic activity. A substantial
increase in the solar wind dynamic pressure early on December 31 was followed by a strong
southward IMF that persisted almost without interruption from 11:00 UT on December 31, 2015
until 09:00 UT on January 1, 2016 (not shown). Figure 1 shows geomagnetic activity indices
Dst, Kp and AE obtained from the OMNI database for the three day interval from 12:00 UT on
December 30, 2015 to 24:00:00 UT on January 1, 2016. A strong electrojet with AE index
greater than 2100 nT at 12:36 UT on December 31, 2015 was followed by two moderate
substorms that enhanced AE at ~14:00 and 18:45 UT on January 1, 2016. The Dst index
responded by reaching a value as low as -110 nT at 00:30 UT on January 1, 2016. Shading
highlights the interval from ~18:55 to 23:02 UT late in the recovery phase and late in the day on
January 1, 2016 when the Van Allen Probes and GOES spacecraft observed the strong
compressional Pc5 pulsations of interest to this study.
The latter interval was marked by strong variations in the solar wind dynamic pressure.
Figure 2 presents Wind observations of the magnetic field and plasma from 16:00 to 24:00 UT
on January 1, 2016, during which time the spacecraft moved from GSM (X, Y, Z) = (194.7, 20.1,
-12.5) Re to (194.8, 23.6, -7.4) Re. Shading marks an interval of depressed magnetic field
strengths and generally anticorrelated enhanced densities, velocities and solar wind dynamic
pressures. The cone angle was less than $45^0$ during this interval. The magnetic field was briefly
aligned with the Sun-Earth line (Bx) at the center of the interval from 20:00-21:00 UT. For most
of the ~4h long shaded interval, IMF Bx (By) was predominantly positive (negative) and the Bz
component remained almost constant near 0 nT, indicating a spiral and equatorial IMF
configuration.
Figure 3 presents RBSP-A and -B and G-13 (MLT~ UT-5) and -15 (MLT~ UT- 9)
trajectories from 15:00 UT to 24:00 UT on January 1, 2016 in the X-Y and X-Z GSM planes.
Open circles mark the beginning of the spacecraft trajectories which are duskward for the GOES
spacecraft and duskward at apogee for the Van Allen Probes. All of the spacecraft were north of
the equator when in the dayside magnetosphere. The thick line segments (dots) indicate the
locations of the spacecraft at the times when (weak) Pc5 magnetic field pulsations occurred.
Figure 4 compares lagged Wind solar wind dynamic pressure variations with G-13 and -
15 observations of the dayside magnetospheric magnetic field. The arrows connect
enhancements of the solar wind dynamic pressure to corresponding compressions of the
magnetosphere. It is relatively easy to associate the GOES magnetic field enhancements with



corresponding features in the solar wind dynamic pressure at the beginning and the end of the
interval but less easy from 19:50 UT to 21:20 UT corresponding to ~ 20:45 UT and 22:15 UT at
the GOES spacecraft. The lag time from Wind to the Earth is not uniform and depends on IMF
orientation. At the beginning and end of the interval, when the IMF was spiral (Bx >0, By <0),
the lag was in the range of ~46 to 58 min. Consistent with expectations, the lag became greater
for the interval from ~ 19:50 UT to 21:20 when the IMF was nearly radial (By and Bz ~0 nT).
The reasonable correspondence of the magnetosphere compressions to solar wind dynamic
pressure variations demonstrates that Wind was a good monitor for solar wind conditions and
that a series of pressure enhancements were applied to the magnetosphere during the interval of
interest. Pc5 pulsation amplitudes at G-13 and -15 were greater during the interval of enhanced
solar wind dynamic pressure and magnetospheric magnetic field strengths than they were at
earlier and later times.

**4. Pulsation Observations**
**4.1. Spatial characteristics of Pc5 pulsations**

Consider the spatial extent, temporal, and spectral characteristics of the compressional

Pc5 pulsations. Figure 5 shows RBSP-A (a) and -B (b) magnetic field observations in GSM
coordinates from 18:40 UT to 21:10 UT and from 20:40 UT to 23:10 UT, respectively, on
January 1, 2016. Taken together, the RBSP-A and -B  observed compressional Pc5 pulsations
that occupied the inner dayside magnetosphere from 5.26 to 5.75 $R_E$ and from 09:56 to 12:44
MLT. Prior to the arrival of the strong solar wind dynamic pressure variations, RBSP-A
observed very weak compressional pulsations with Pc5 periods and amplitudes of 1-3 nT from
18:15 to 18:55 UT. After G-15 began to observe compressions at about 19:00 UT (Figure 4), the
amplitudes of the pulsations at RBSP-A began to increase (Figure 5). They increased
prominently to values ranging from 10 to 15 nT in the Bz component with the peak amplitudes
occurring prior to local noon. The Bz component oscillated out of phase with the Bx component
and in quadrature with the By component. The compressional pulsations at RBSP-A ended at
20:58 UT. RBSP-B observed similar compressional Pc5 pulsations from 20:46 UT that ceased
simultaneously with the end of the magnetospheric compression seen by G-15 about 23:02 UT.

Figure 6 shows G-13 and -15 observations of the magnetic field in GSM coordinates

from 18:00 UT to 24:00 UT. The spacecraft observed long-duration compressional Pc5
pulsations over a wide longitudinal region in the pre- and post-noon magnetosphere from 10:00
to 15:20 MLT (Figure 3). There was also weak preexisting Pc5 wave activity before the strong
solar wind dynamic pressure variations. G-15 observed pulsations with amplitude less than 5 nT





from 18:28 UT to 19:04 UT.  Then after the subsequent magnetospheric compressions their
amplitude increased to values ranging from 10 to 16 nT with peak amplitudes prior to local noon
as for the Pc5 pulsations observed by RBSP-A and -B.  G-13 observed weak Pc5 pulsations with
amplitudes of 2-4 nT throughout most of the time interval from 16:40 UT to 21:00 UT.  During
the    interval    from    19:34    UT    (~14:45    MLT)    to    20:10    UT
(~15:20 MLT), the pulsations reached slightly stronger amplitudes of 5-8 nT.  At 23:02 UT all
wave activity observed at GOES stopped.
We converted the magnetic field observations from GSE into field-aligned coordinates
(FAC). Here the Z axis lies parallel to the locally-averaged magnetic field. The Y axis points
approximately azimuthally eastward and is transverse to B and to the outward radius vector. The
X axis completes the right-handed system and is directed approximately radially outward from
Earth. Figure 7 presents RBSP-A and -B magnetic field observations in FAC.   The Bz
component reached 15 nT and it is the strongest one as is characteristic of compressional
pulsations. The amplitudes of the Bx and Bz components are weaker than those of the Bz
component and did not exceed 7 nT.  Simultaneous RBSP-A and -B electric and magnetic field
measurements provide an opportunity to study the structure of the Pc5 waves.  Determining the
harmonic mode of the Pc5 waves requires us to consider the phase of the azimuthal component
of the electric field Ey with respect to the radial component of the magnetic field Bx as a
function of latitude [Takahashi et al., 2011].  Figure 8 shows that the phase of the Ey component
leads that of the Bx component by 90° at RBSP-A from 19:10 UT to 20:00 UT and therefore the
Pc5 waves are second harmonic in nature.

**4.2. Spectral characteristics**
We calculated dynamic spectra for the magnetic field pulsations.  Figure 9 presents the
radial, azimuthal and compressional components of the dynamic spectra of the magnetic field at
RBSP-A and -B from 18:00 to 21:10 UT and from 20:00 UT to 23:10 UT on January 1, 2016,
respectively.  The color bar on the right shows the scale for power for frequencies ranging from
0 to 41 mHz in each component.  The magnetic field exhibited several wide-band enhancements
at frequencies ranging from 4 to 29 mHz.  As expected for compressional Pc5 pulsations, both
spacecraft observed the strongest power densities in the Bz component at dominant frequencies
of ~4.5-6 mHz.  Red arrows in the Bz panels of Figure 9 for RBSP-A and -B  indicate the double
frequency pulsations at ~5.5 mHz and ~11 mHz.  We calculated Fourier spectra for the three
components of the RBSP-A and -B magnetic field in 600 second sliding-averaged mean FAC for
each thirty min interval during the event.  Figure 10 presents examples of Fourier spectra
calculated for the RBSP-A and -B magnetic field from 19:30 UT to 20:00 UT and from 22:30
UT to 23:00 UT, respectively, on January 1, 2016.    The red arrows show the dominant
frequencies at 5.5 and 5 mHz observed at the two spacecraft, corresponding to periods of 170-
200 s.  RBSP-A and -B were situated three hours in local time apart, the similar frequencies
indicate that conditions in the dayside magnetosphere remained steady for a long time and over a
broad region.

In passing, we note the presence of Pc4 pulsations. Returning to Figure 9, we see enhanced

power densities at frequencies of ~22-29 mHz with dominant frequencies from 23 to 27 mHz
primarily in the radial Bx component.   These can be ascribed to poloidal Pc4 produced
simultaneously with the Pc5 but likely with another energy source.  The frequencies of the Pc4
pulsations decrease with increasing radial distance, as expected for resonant standing Alfvén
waves [Sugiura and Wilson, 1964].   Pulsation periods depend upon the magnetic field line
length, the magnetic field magnitude, and the ion density.   Shorter field line lengths and
enhanced magnetic field strengths closer to Earth decrease pulsation periods.   Blue arrows in
Figure 9 indicate Pc4 pulsations at ~25-27 mHz.

Figure 11 presents dynamic spectra for the G-13 and -15 magnetic field in FAC from 18:00

UT to 24:00 UT on January 1, 2016.  Spectral power was calculated for frequencies from 0 to 48
mHz.    Like the RBSP-A and -B  magnetic field spectra, there are two broad frequency band
enhancements corresponding to Pc4 and 5 frequencies.   The dominant frequencies for the
compressional Pc5 pulsations occur from 4.5 to 6.5 mHz.  These frequencies are similar to those
observed by Van Allen Probes and we suppose that they were generated by the same sources.
The Pc4 pulsations are most pronounced in the radial Bx component and display strongest
spectral power densities in the frequency range from 13 to 21 mHz.  These frequencies are lower
than those observed by Van Allen Probes, as expected since the GOES spacecraft were located
further radially outward from Earth [Sugiura and Wilson, 1964].  The frequencies of the long-
lasting Pc4 pulsations observed by G-15 depended on local time.  They decreased from 20-22
mHz in the prenoon magnetosphere to 14-17 mHz near local noon, perhaps in response to
differing conditions (e.g., densities).  Takahashi at el. [1984] noted that an increase in plasma
mass density from morning to afternoon is typical at geosynchronous orbit.   Since the
frequencies of the Pc4 pulsations depended on local time and radial distance from Earth, their
sources must be more localized than those for the Pc5 pulsations.





### 4.3. Particle signatures

Energetic particle observations provide further information concerning this event. We inspected RBSP-A and -B MagEIS observations of energetic particles from 18:30 UT to 21:00 UT and from 20:40 UT to 23:10 UT on January 1, 2016, respectively, and found that the intensities of electrons with energies from tens of keV to 2 MeV oscillated with Pc5 periods corresponding to those of the magnetic field. Figure 12 shows these oscillations. The energetic electron fluxes oscillated out of phase with the magnetic compressional component of Pc5 pulsations and did not display any phase differences across all energies. The depth of modulation (the peak to valley ratio) is larger for higher energy electrons consistent with the results of Liu et al. [2016] who interpreted similar observations in terms of mirror mode waves. Kivelson and Southwood [1985] noted that the maintenance of pressure balance in low-frequency compressional waves usually requires the presence of some pitch angle anisotropy and the antiphase relation between P and B suggests that particle pitch angle distributions peak near 90°. Figure 13 presents RBSP-A and -B observations of pitch angle distributions for electrons with energies from 54 keV to 1060 keV from 18:30 to 21:00 UT and from 20:40 UT to 23:10 UT on January 1, 2016, respectively. The figure confirms that pitch angle distributions. peak near 90°. Furthermore, it shows that the electron intensities display quasi-periodic enhancements at all energies with the strongest at pitch angles near 90º.

### 4.4. Double-frequency pulsations

When RBSP-A and -B and G-15 were in the vicinity of the geomagnetic equator the compressional Pc5 pulsations displayed peculiar features indicating frequency doubling. Here the compressional components oscillated with a frequency double that for the transverse components. Coleman [1970] was the first to report observations of such events in the geosynchronous magnetic field. Higuchi et al. [1986] called them harmonic structures when the first and second harmonics exhibited similar amplitudes and transitional structures when the amplitudes of the alternating peak were different. Takahashi [1987b] interpreted double-frequency oscillations in terms of a model invoking the second harmonic structure of an antisymmetric standing wave in which the location of the equatorial node of field-lined displacement oscillates in phase with the wave. Cheng and Qian [1994] presented a model for the magnetic field perturbations during the pulsations reported by Takahashi et al. [1987a, 1990]. Figure 6 in the paper of Korotova et al. [2013] illustrates how low-latitude spacecraft can observe two magnetic field strength enhancements per wave cycle when the equatorial node oscillates up and down in phase with an antisymmetric compressional wave. Right at the equator





the spacecraft observes identical amplitudes for the two compressions. At any other latitude the
two compressions at the spacecraft will have different magnitudes and the imbalance between
them increases when the spacecraft moves farther from the equator. Takahashi et al. [1997b]
showed that that a latitudinal shift of a fraction of degree can turn the harmonic structure of Bz
into nonharmonic. Spacecraft located from the magnetic equator at a large distance do not
observe frequency doubling, just a single enhancement. Korotova et al. [2013] derived the
latitudinal structure of the waves by invoking north-south sloshings of the low-latitude node.
Figure 14 presents (a) RBSP-A and -B observations of double frequency magnetic
pulsations and (b) their locations in the X-Y GSM and X-Z SM planes. Dashed lines in Figure
14 indicate intervals when the double frequency pulsations in Bz are most prominent: 20:45-
20:54 UT at RBSP-A and 21:03 UT to 21:31 UT at RBSP-B. However, the amplitudes of the
second harmonic are generally much lower than those of the first harmonic. At these times, e.g.
from 20:05 to 20:45 UT at RBSP-A and 21:35-21:55 UT at RBSP-B, the second harmonic
compressions in Bz are barely perceptible. Model predictions for the magnetic field
perturbations associated with an equatorial node whose latitude oscillates in phase with an
antisymmetric poloidal wave indicate that the ratio of the amplitudes of the first to second
harmonic compressions should change with latitude, being ~1 at the average position of the low-
latitude node and ~0 at and beyond the maximum latitude to which the oscillating node can reach
[Takahashi et al., 1987b]. To determine the meridional motion of the magnetic field node we
measured amplitudes of the first and second harmonics of the compressional pulsations. We
found that RBSP-A observed ratios near 1 at $Z_{SM}$ = ~0.08 Re while RBSP-B observed ratios near
1 at $Z_{SM}$ = ~0.10 Re. These are the locations where the southward-moving spacecraft pass
through the mean positions of the equatorial node. Figure 14a shows that RBSP-A observed
second harmonics from $Z_{SM}$ = 0.25 to 0.04 Re, while RBSP-B observed them from $Z_{SM}$ = 0.19 to
-0.08 Re. Consequently, we believe that the equatorial node oscillated with an amplitude of at
least 0.15 to 0.18 Re. Note however, that the ratio of the first to second harmonics does not
show a smooth transition as the spacecraft move equatorward. Either the amplitude of the
compressional pulsation or the meridional oscillation in the equatorial node varied in time,
probably abruptly.
Figure 12 shows that the compressional pulsations modulated energetic electrons
observed by RBSP- A and - B and we should therefore expect to find the signatures of the
double-frequency pulsations not only in the magnetic field but also in the fluxes of particles.
Takahashi et al. [1990] reported AMPTE/CCE observations of compressional Pc5 pulsations that
demonstrated harmonically related transverse and compressional magnetic oscillations that



modulated the flux of medium energy protons (E > 10 keV) with double frequency but did not
discuss the event in detail. We report the first evidence for meridional sloshing of the equatorial
node in the simultaneous compressional Pc5 pulsations and variations of electrons fluxes and
electron densities observed by MagEIS and Hope, respectively. Figure 15 presents RBSP-A (left
panel) and -B (right panel) electron fluxes for energies at 31.9 keV and 54.8 keV, electron
densities and the Bz component of the magnetic field in FAC from 19:00 UT to 21:00 UT and at
RBSP-B from 20:46 UT to 22:10 UT. The panels in the bottom of Figure 15 present expanded
views of 20 min intervals with the double-frequency pulsations. The Bz component of the
magnetic field varies with double frequencies out of phase with the fluxes of electrons and
densities. This study gives better insight into the nodal structure of the waves and helps to
clarify their source.

**4.5. Testing Pc4-5 pulsation generation mechanisms**
We tested several causes for the Pc4-5 pulsations, including solar wind pressure pulses,
the KH instability on the magnetopause, drift-bounce resonant particle interactions, and the
mirror-mode instability. First, with the exception of the interval from 19:35 UT to 19:55 UT, the
Wind observations shown in Figure 2 provide no evidence for periodic solar wind drivers in the
Pc5 range, be they density variations or IMF fluctuations, thus ruling out solar wind pressure
pulses as the direct cause of the Pc4-5 pulsations. We then considered the possibility of KH
waves. These waves are expected when the solar wind velocity is high and both the
magnetosheath and magnetospheric magnetic fields lie transverse to the magnetosheath flow, i.e.
on the flanks of the magnetosphere when the IMF points southward or in particular northward
[e.g., Guo et al., 2010]. As shown in Figure 2, the solar wind velocity during the interval when
the Pc5 events occurred was only moderate, 400-450 km/s. Furthermore, the IMF did not point
either strongly northward or southward. Therefore, we conclude like many previous researchers
that the compressional Pc5 pulsations were excited by processes internal to the magnetosphere.
Southwood [1981] and Kivelson and Southwood [1985] described how the resonant drift-
bounce interaction of particles with an azimuthally-propagating wave generates large amplitude
ULF waves in an inhomogeneous background field. For this to happen, the wave frequency $\omega$
must satisfy the resonance condition:
$$\omega - m\omega_d - N\omega_b = 0, \qquad (1)$$
where $\omega_d$ and $\omega_b$ are the angular drift and bounce frequencies, N is an integer, and m is the
azimuthal wave number. Southwood [1973] predicted that particle flux oscillations just above



and below   the resonant energy should be 180° out of phase. As Figure 12 demonstrates, RBSP-
A and -B  ion and  below      electron observations provide no evidence for any such phase
reversal at any relevant energy.   We exclude the drift-bounce resonance as the cause of the
compressional Pc5 pulsations.

Finally, we examined the mirror instability criterion.  The mirror instability is a kinetic

phenomenon that occurs spontaneously in anisotropic high β plasmas when the ratio of
perpendicular to parallel pressures is large [Southwood and Kivelson, 1993].  The test for the
mirror instability is approximately:

$\Gamma = 1 + \beta_\perp [1 - T_\perp / T_\parallel] < 0,$          (2)

where $T_{\parallel, \perp}$ are the plasma temperatures parallel and perpendicular to the ambient magnetic
field and $\beta_\perp$ is the ratio of the perpendicular component of the thermal plasma pressure to the
magnetic pressure.  For our calculations we obtained the magnetic field data from EMFISIS and
thermal plasma pressures perpendicular and parallel to the magnetic field from RBSPICE.  We
used the density and temperature from HOPE to calculate the parallel and perpendicular thermal
pressures within the energy range covered by this instrument, but found these pressures to be
small compared to those from RBSPICE.   Consequently, our calculations neglect the
contributions from HOPE to the thermal pressures.

Figures 16a and b show RBSP-A and -B plasma and magnetic field parameters

characterizing the pulsations. The upper panels indicate that magnetic field and plasma pressures
vary in antiphase during the Pc5 pulsations.  However, the total pressure is not balanced as might
be expected for mirror mode waves.  We suppose that this is because the RBSPICE (or even the
RBSPICE + HOPE) plasma instruments do not observe the entire plasma distribution.  Assuming
that the total plasma pressure is proportional to the fraction that RBSPICE does observe, we
scaled the thermal plasma pressures observed by RBSPICE upward to values that cause the sum
of the magnetic and perpendicular thermal plasma pressure variations associated with the waves
to be approximately constant during the intervals from 19:03 UT to 19:14 UT for RBSP-A and
from 22:32 UT  to 22:56 UT for RBSP-B.  The upward scaling factors were 1.97 and 1.69,
respectively.  We then applied these factors to both the perpendicular and parallel pressures.  The
third panels of Figures 16a and b show the values of $\beta_\perp$ calculated from these scaled pressures.
Shaded grey areas in the fourth panels show when the drift mirror instability is satisfied ($< 0$).
As the test for the mirror instability is satisfied throughout most of the intervals of enhanced
temperature (pressure) anisotropy and $\beta > 1$ at RBSP-A and -B, we attribute the compressional
Pc5 pulsations observed on January 1, 2016 to the mirror instability.



**Conclusions**



403   We used Van Allen Probes and GOES multipoint magnetic field, electric field,  plasma

and energetic particle observations to study the nature of compressional Pc5 pulsations at the end
of a strong magnetic storm on January 1, 2016.  From ~ 19:00 UT to 23:02 UT the
magnetosphere was compressed and transient increases of the total magnetic field strength
occurred every 20-40 min. During this interval the spacecraft observed compressional Pc 5
pulsations over a large longitudinal extent.  They occupied the dayside magnetosphere from 5.26
to 6.6 Re and from 09:56 to 15:20 MLT. The subsequent    solar wind pressure increases and
magnetospheric compressions enhanced the amplitude of Pc5 wave activity to values from 10 to
16 nT.  The strongest amplitudes occurred prior to local noon.  They were observed w h e n  t h e
I M F  cone angle was less than $45^{O}$.  We studied the wave mode of the Pc5 pulsations and found
that they had an antisymmetric structure.

414   The greatest spectral power densities observed at RBSP-A and -B  occurred in the

north/south, or Bz, component of the magnetic field at frequencies of ~4.5-6.0 mHz.  The two
spacecraft observed similar frequencies, indicating that conditions within the dayside
magnetosphere remained steady for a long time and over a broad region.  Enhanced spectral
power densities at frequencies of ~22-29 mHz in the radial Bx component can be attributed to
the simultaneous generation of poloidal Pc4 pulsations by a different mechanism.   The
frequencies of the Pc4 pulsations diminished with increasing radial distance.  The dominant
frequencies for the compressional Pc5 pulsations observed by GOES resembled those observed
by RBSP-A and -B  and we suppose that they were generated by the same sources. The Pc4
pulsations   displayed   frequencies   that   were
lower than those observed by RBSP-A and -B, as expected since the GOES spacecraft were
located further radially outward from Earth. Since the frequencies of the Pc4 pulsations
depended on local time and radial distance from Earth, their sources must be more localized than
those for the Pc5 pulsations.

428   When the spacecraft were in the vicinity of the geomagnetic equator, RBSP-A observed

meridional sloshing of the equatorial wave node from $Z_{SM}$ = 0.25 to 0.04 Re, while RBSP-B
observed them from $Z_{SM}$ = 0.19 to -0.08 Re. Consequently, we believe that the motion of the
meridional oscillation of the position of the equatorial node was at least 0.15 to 0.18 Re. We
found that RBSP-A observed ratios near 1 at $Z_{SM}$ = ~0.08 Re while RBSP-B observed ratios near
1 at $Z_{SM}$ = ~0.10 Re.  These were the locations where the southward-moving spacecraft RBSP-A
and -B passed through the mean positions of the equatorial node at $Z_{SM}$ = ~0.08 Re and at $Z_{SM}$ =





~0.10 Re, respectively.  We report the first evidence for meridional sloshing of the equatorial
node in the double-frequency variations of electrons fluxes and electron density observed by
MagEIS and HOPE, respectively.

The energetic particles observed by RBSP-A and -B  exhibited  a regular periodicity over a

broad range of energies from tens of eV to 2 MeV with periods corresponding to those of the
compressional component of the ULF magnetic field. The electron intensities exhibited quasi-
periodic enhancements at all energies with the most intense at pitch angles near 90°. The
energetic electron fluxes oscillated out of phase with the magnetic field and did not display any
phase shift across all energies. The depth of modulation was larger for higher energy electrons.
We searched for possible solar wind triggers and discussed generation mechanisms for the
compressional Pc5 pulsations in terms of drift mirror instability and drift bounce resonance. We
interpret the compressional Pc5 waves in terms of drift-mirror instability.



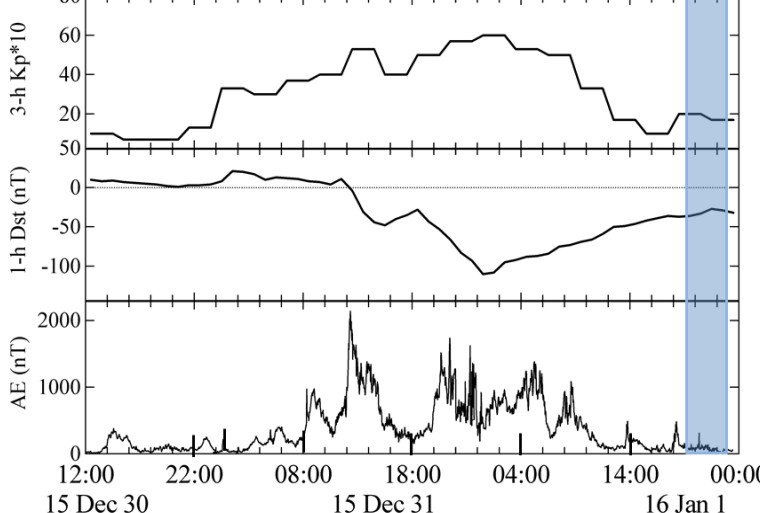


Figure 1. Geomagnetic activity indices  Kp, Dst and AE indices from 12:00 UT on December 30
to January 2, 2016 available from the OMNI database ( http://omniweb.gsfc.nasa.gov).    The
shading highlights the interval when the spacecraft observed Pc5 compressional  pulsations.


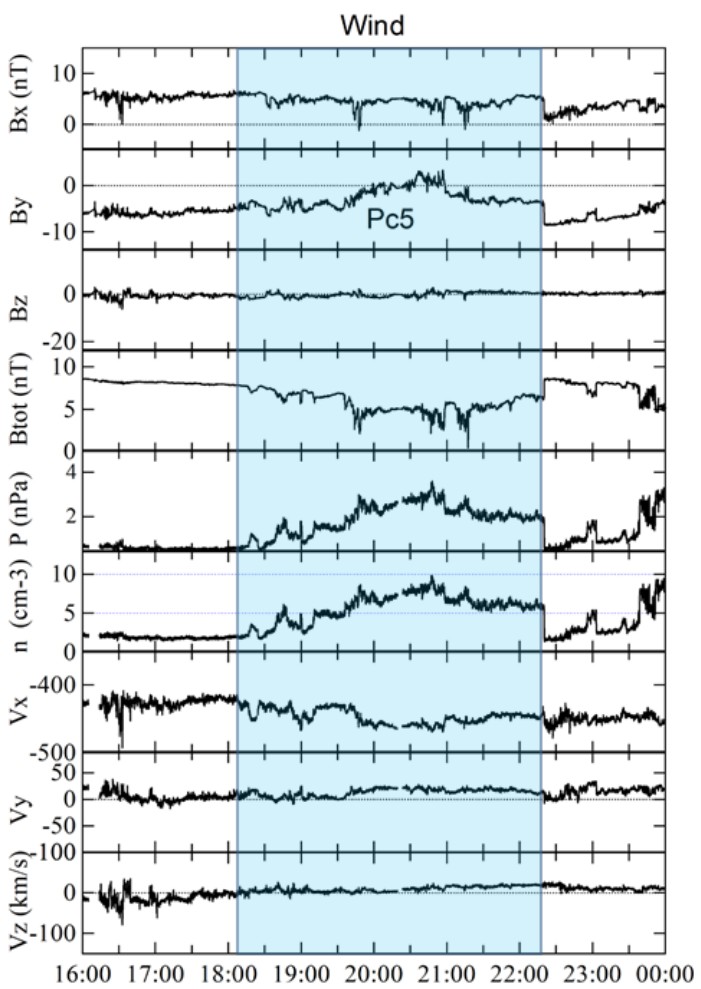


Figure 2. Wind observations of the magnetic field and plasma from 16:00 UT to 24:00 UT on
January 1. 2016. Shading highlights the interval of interest.

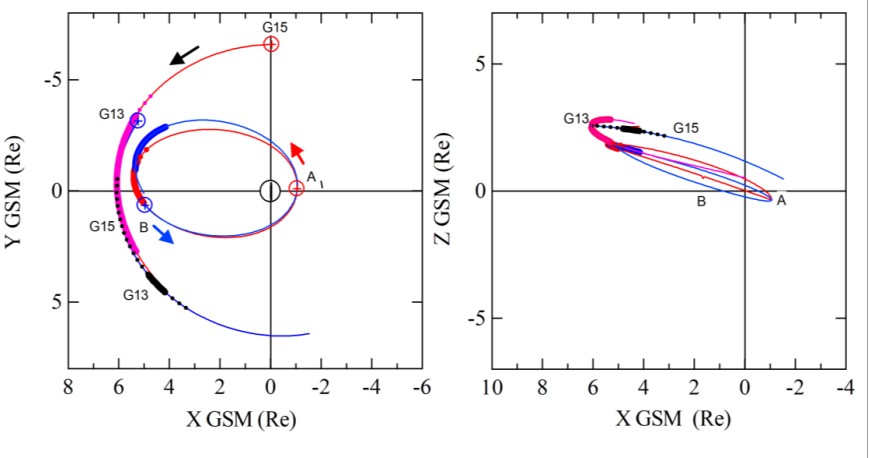


Figure 3. Trajectories of RBSP-A (red) and -B (blue) and G-13 (blue) and -15 (red) from 15:00
UT to 24:00 UT on January 1, 2016 in the X-Y and X-Z GSM planes. Open circles mark the
beginning of the spacecraft trajectories, that are duskward in the dayside magnetosphere. The
thick line segments indicate the locations of the spacecraft at the times when compressional Pc5
magnetic field pulsations occurred. Dots mark their locations where weak pulsations (A < 5 nT)
occurred.

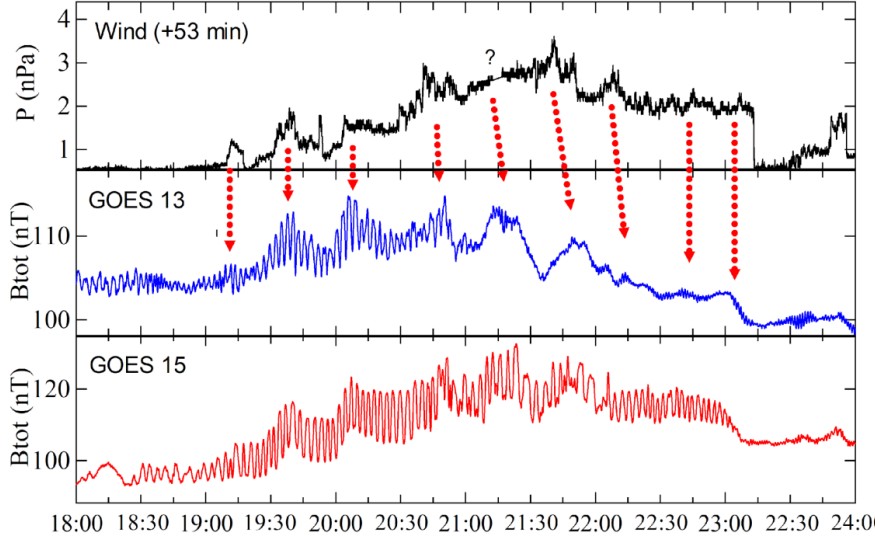


Figure 4. Observations of the solar wind dynamic pressure at Wind (time shifted) and the total
magnetic field strength at G-13 and -15 from 18:00 UT to 24:00 UT. The arrows connect



enhancements of the solar wind dynamic pressure to corresponding compressions of the
magnetosphere.

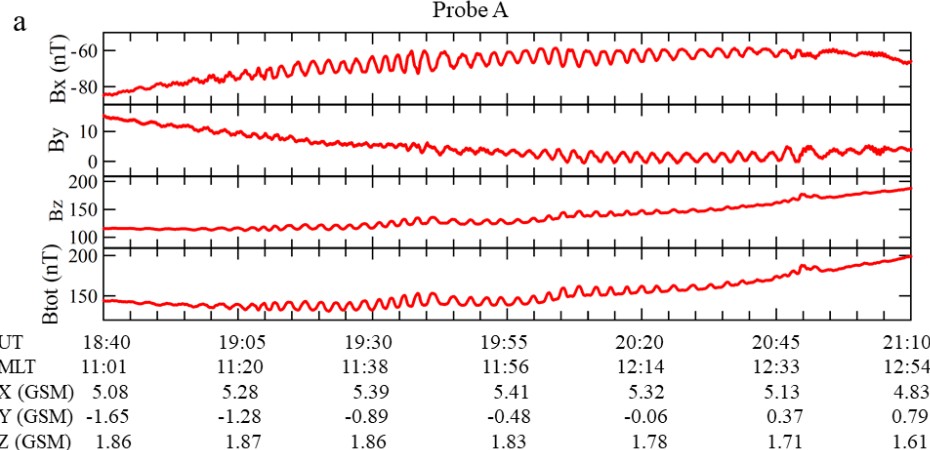


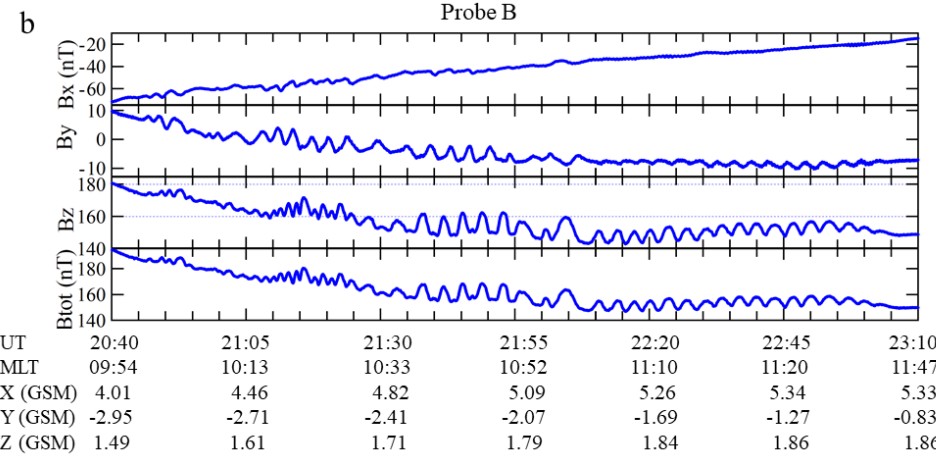


Figure 5. RBSP-A (a) and -B (b) magnetic field observations in GSM coordinates from 18:40
UT to 21:10 UT and from 20:40 UT to 23:10 UT on January 1, 2016, respectively. Beneath the
panels are listed the universal time (UT), magnetic local time (MLT), X (GSM). Y (GSM) and Z
(GSM) in Earth radii.





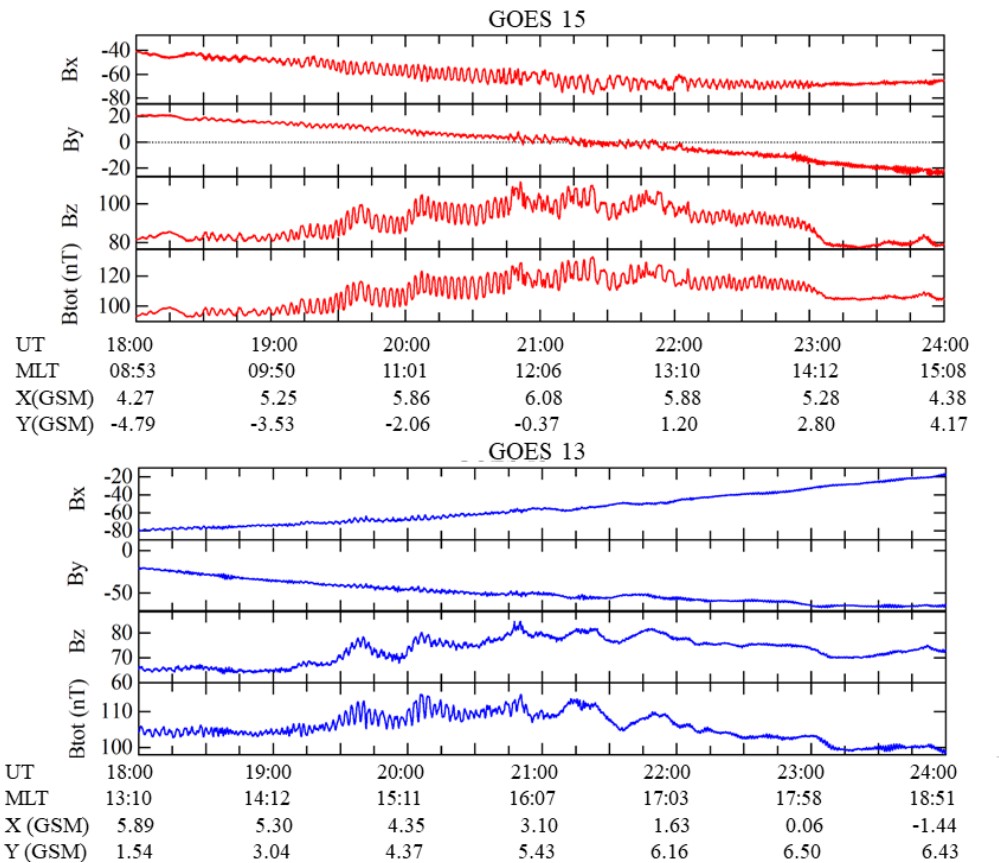

Figure 6.  G-13 and 15 observations of the magnetic field in GSM coordinates from 18:00 UT to 24:00 UT on January 1, 2016. Beneath the panels are listed the universal time (UT), magnetic local time (MLT in SM), X (GSM) and Y (GSM) in Earth radii.



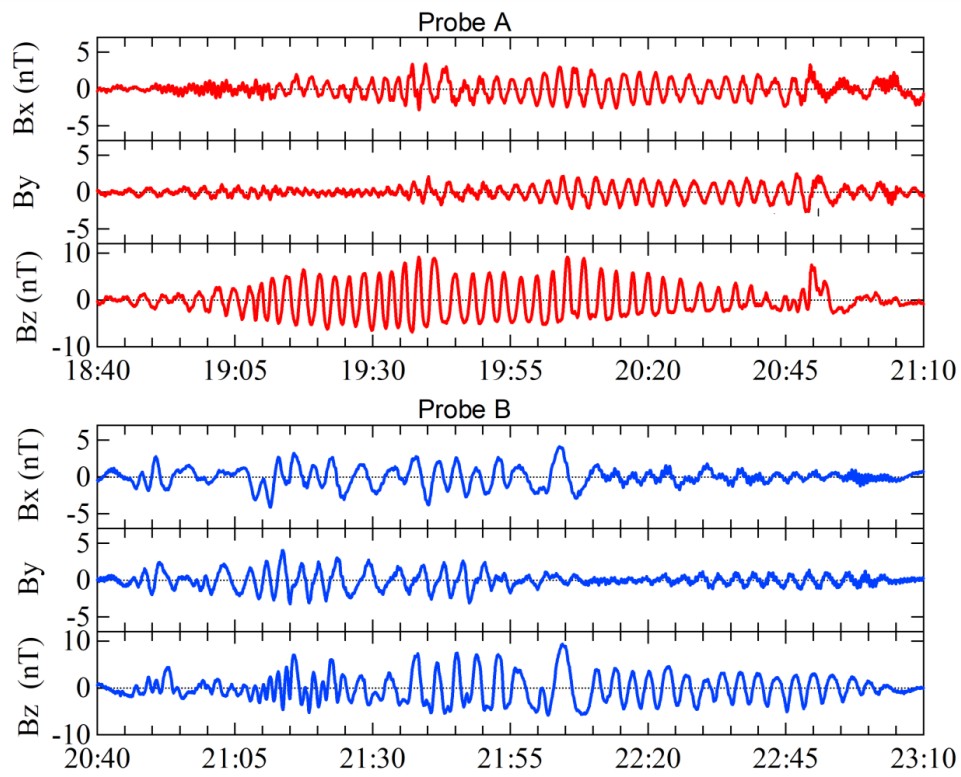

Figure 7.  RBSP-A  and -B  magnetic field observations in field-aligned coordinates from 18:40
UT to 21:10 UT and from 20:40 UT to 23:10 UT on January 1, 2016, respectively.

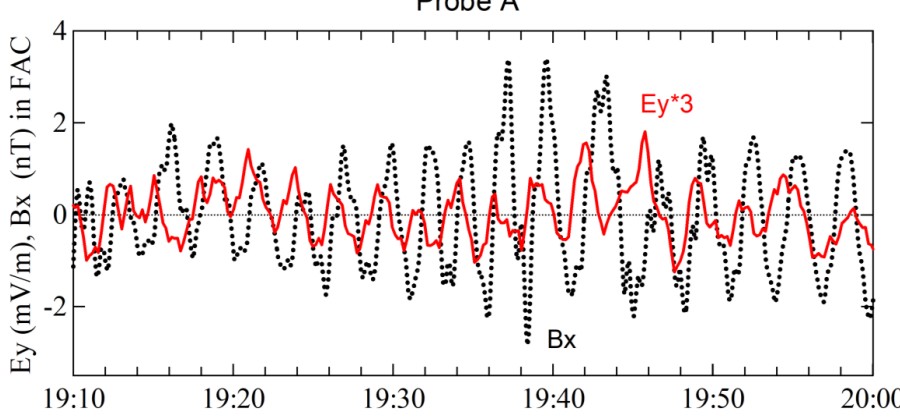

Figure 8. The phase difference between the RBSP-A azimuthal component of the electric field
(red curve is boxcar smoothed) and the radial component of the magnetic field Bx  in field-
aligned coordinates (dashed curve) from 19:10 UT to 20:00 UT on January 1, 2016. The
amplitude of Ey was multiplied by a factor of 3 to better display the visual effects.

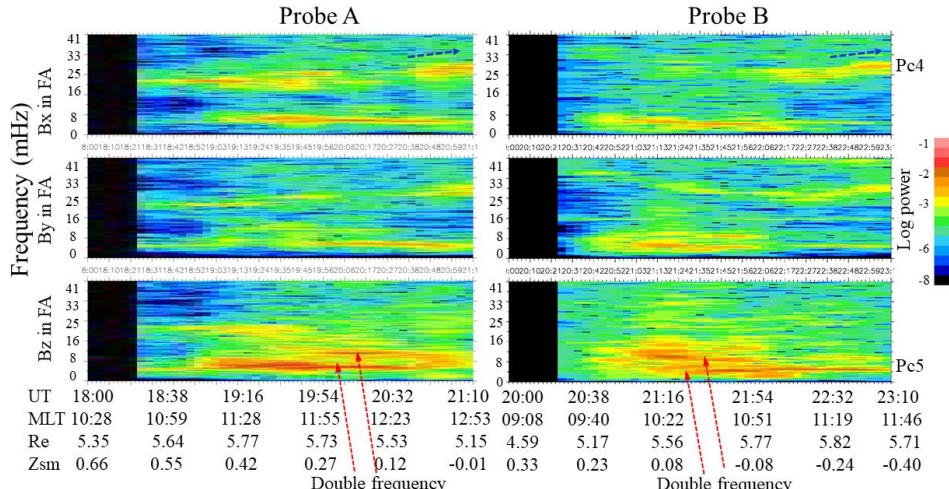


Figure 9. Three component dynamic spectra of magnetic field data at RBSP-A and -B from
18:00 to 21:10 UT and from 20:00 UT to 23:10 UT on January 1, 2016, respectively. Beneath
the panels are listed the universal time (UT), magnetic local time (MLT), radius (Re) and Z
(SM) in Earth radius.

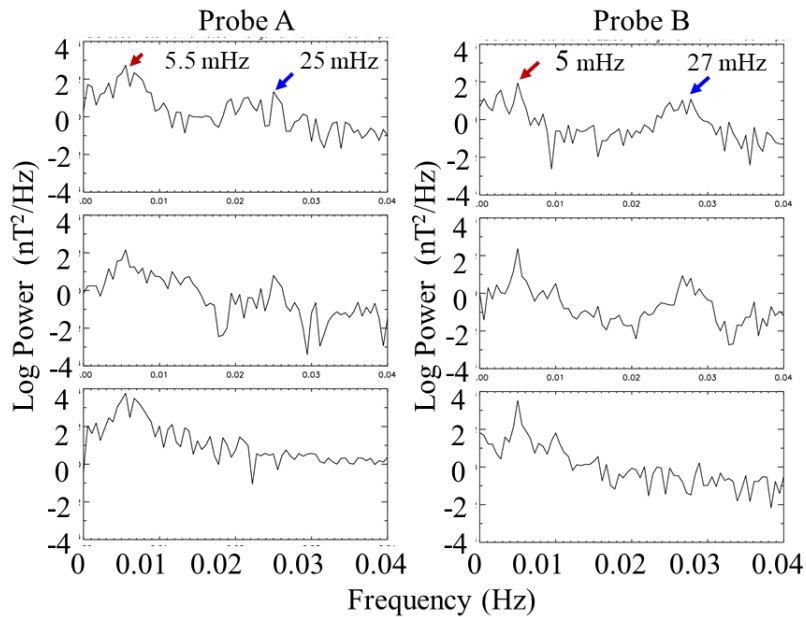






Figure 10. Fourier spectra calculated for the radial, azimuthal and compressional components of
the RBSP-A and -B magnetic field in 5-minute sliding averaged mean field-aligned coordinates
from 19:30 UT to 20:00 UT and from 22:30 UT to 23:00 UT on 1 January, 2016.

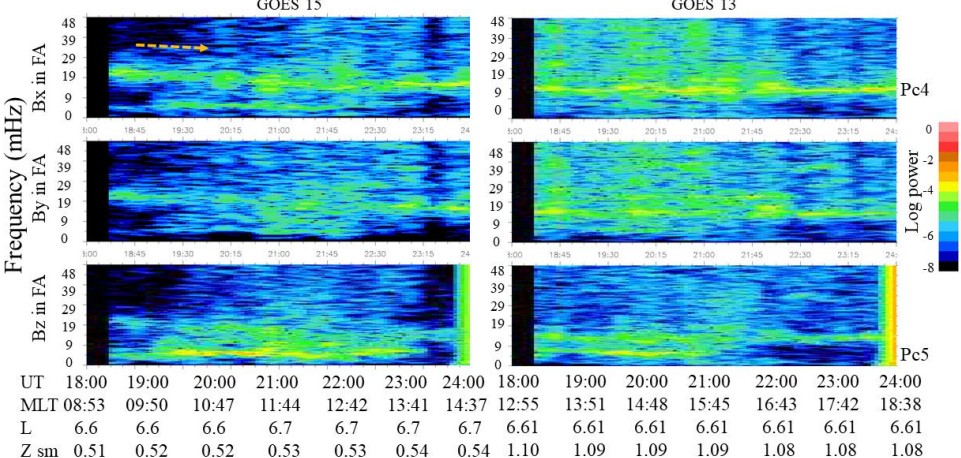


Figure 11. Three components of dynamic spectra of  the magnetic field data at  G-15 and  G-13
from 18:00  UT to 24:00 UT  on January 1, 2016.  Beneath the panels are listed the universal
time (UT),  magnetic local time (MLT in SM), L  and Z (SM) in Earth radii.

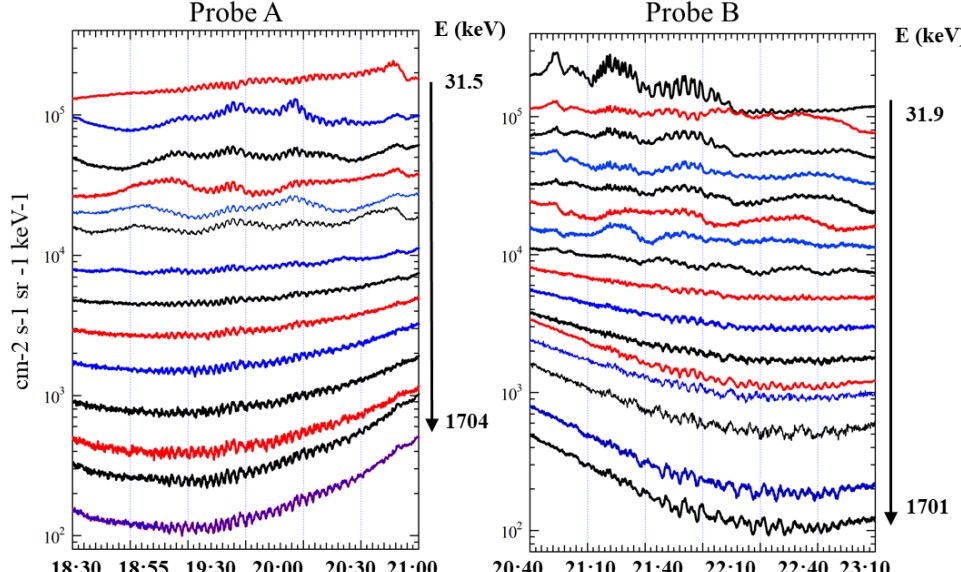






Figure 12. RBSP-A and -B observations of electron fluxes in the range of energies from 31.5
keV to 1704 keV  from 18:30 UT to 21:00 UT and from 20:40 UT to 23:10 UT, respectively.

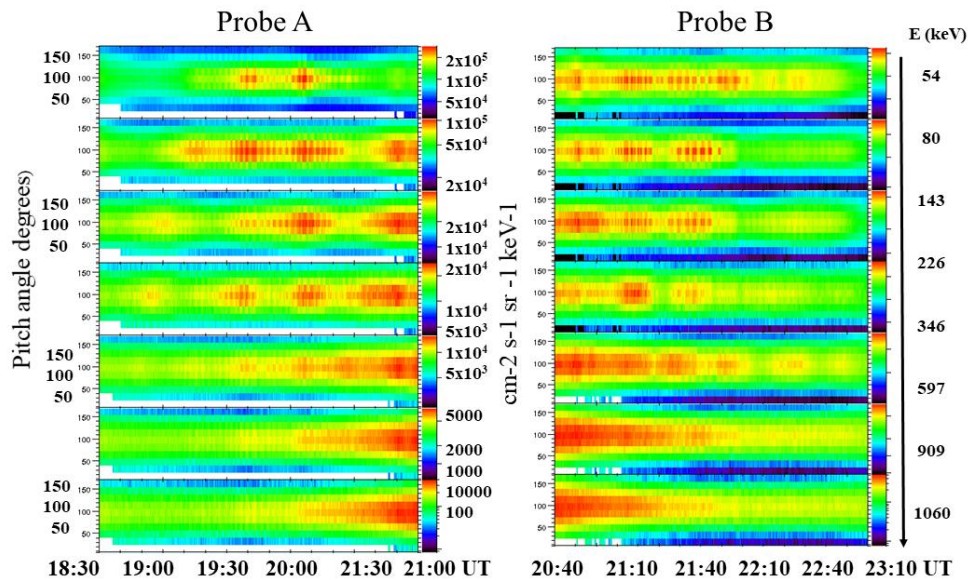


Figure 13. RBSP-A and -B observations of pitch-angle distributions for electrons in the  range of
energies from  54 keV and 1060 keV  from 18:30 to 21:00 UT and from  20:40 UT to 23:10 UT
on January 1, 2016, respectively.

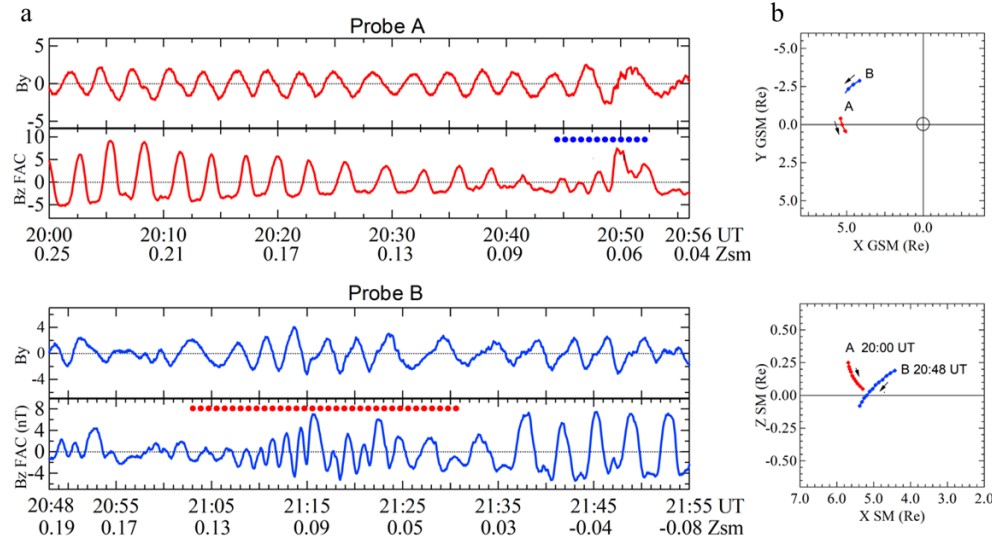




Figure 14. RBSP-A and -B observations of double frequency pulsations (a) from 20:00 UT to 20:56 UT and from 20:48 UT to 21:55 UT, respectively, and (b) their locations in the X - Y GSM and X - Z SM planes. Red and blue dashed lines mark the intervals with harmonic structure of double-frequency pulsations.

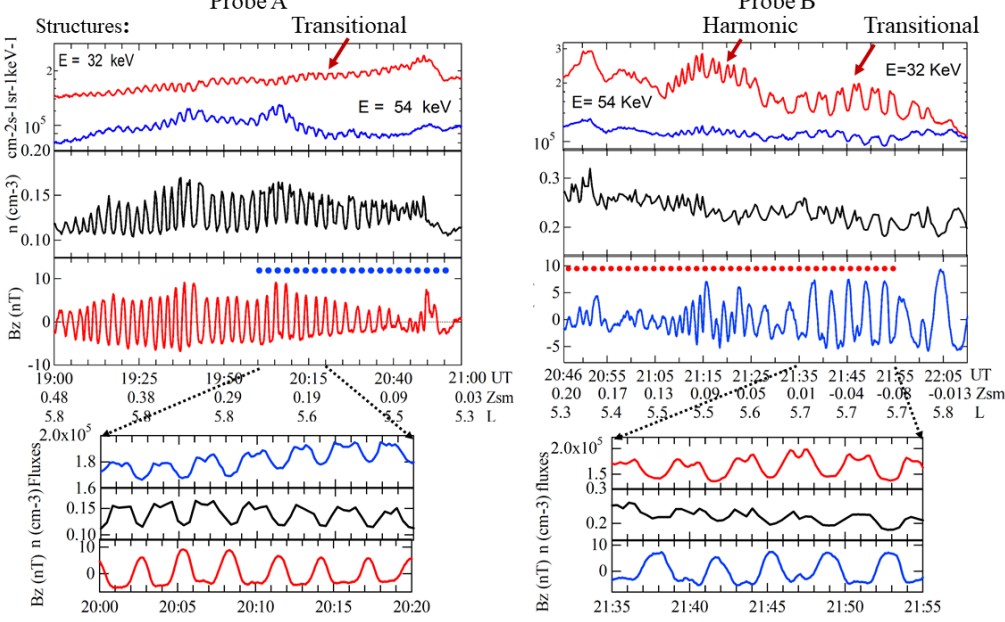

Figure 15. RBSP-A (left panel) and -B (right panel) presents electron fluxes for energies at 31.9 keV and 54.8 keV from EMFISIS, electron densities from HOPE and the Bz component of the magnetic field in field-aligned coordinates from MagEIS from 19:00 UT to 21:00 UT and from 20:46 UT to 22:10 UT, respectively. Dashed lines mark the intervals of observations of double-frequency pulsations. The panels in the bottom of the figure present expanded views of 20 min intervals with the double-frequency pulsations to better visualize their features.



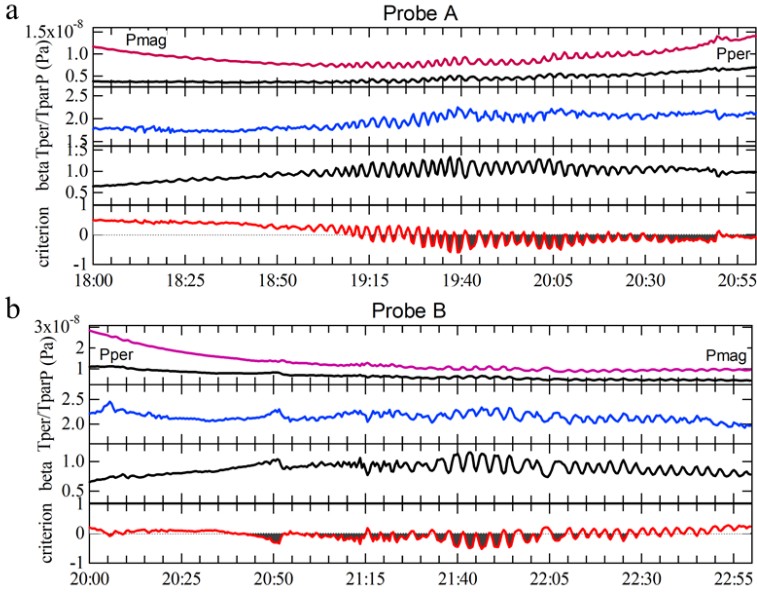


Figures 16a and b. RBSP-A and -B plasma and magnetic field parameters characterizing the
pulsations. From top to bottom, the figure shows the magnetic pressure, perpendicular plasma
pressure, the ratio of the plasma temperatures perpendicular and parallel to the magnetic field,
beta, and the results for the mirror instability criterion on January 1, 2016. Shaded grey areas
indicate the times when the drift mirror instability is satisfied (< 1).
**Data availability**. Data used in the paper are available publicly at
http://cdaweb.gsfc.nasa.gov/istp_public/ (Coordinated Data Analysis Web, NASA, 2018). GOES
data were obtained from http:// satdat.ngdc.noaa.gov/sem/goes/data/new_full/ (NOAA, 2018).
The electric field data were obtained from http://www.space.umn.edu/ rbspefw-data (Wygant and
Breneman, 2017).
**Author contributions**. GK drafted and wrote the paper with participation of all coauthors. DS
conceived ideas, ME, ST, HS, CK –consulting regarding the data analysis, RR – software
development, MB–consulting regarding drift mirror instability test.
**Competing interests**. The authors have no conflict of interest.
**Acknowledgements.** The Van Allen Probes mission is supported by NASA. NASA GSFC's
CDAWEB provided Wind and GOES observations, while SSCWEB provided Van Allen Probes
EPHEMERIS. GK was supported by NASA contract no 80NSSC19K0440. M. A. B. is
grateful to the STFC (grant ST/R000697/1).

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
