# Peer review of "Multipoint Observations of Compressional Pc5 Pulsations in the Dayside"

_Annales Geophysicae, 2020_

## Referee Comment (RC1) · Anonymous Referee #1 · 22 Jul 2020

I have read the manuscript angeo-2020-32 with title "Multipoint Observations of Compressional Pc5 Pulsations in the Dayside Magnetosphere and Corresponding Particle Signatures". I believe that the observational evidence for the generation of Pc5 pulsations due to mirror instability are quite important yet I'm left with the feeling that the importance of the authors' conclusions are somehow lost into many unnecessary details reported. Moreover, the authors have included 16 figures in the manuscript yet some of them (or some panels at least) are not discussed at all or even they do not play an important role in the conclusions. You can find my comments below.

Lines 103-114: Please merge this chapter with introduction.

[Figure]

Lines 136-138: If it is not that important (I guess it is not since it is not shown) it shouldn't be mentioned at all.

Line 152: Define cone angle. Also I don't see it anywhere in the figure. Instead I see the three components of speed which are not discussed at all. Figures are already too many (16!!!). Since Kp and speed are not discussed at all remove them and merge figures 1 and 2 (or even better provide the total solar wind speed only).

Lines 170-178: The authors discuss the time-lag between solar wind pressure and compressions at GEO. Is this really important for the conclusions of this work? Moreover, I'm left with the feeling that the use of GOES measurements, in general, do not provide any significant observational evidence in this work. If I'm wrong then I believe that it should be discussed more clearly but if I'm not the authors should consider not using it at all.

Lines 187-189: The authors state "Prior to the arrival of the strong solar wind dynamic pressure variations, RBSP-A observed very weak compressional pulsations with Pc5 periods and amplitudes of 1-3 nT from 18:15 to 18:55 UT." This is not shown anywhere.

In figures 5 and 6 the authors show the magnetic field in GSM coordinates along with the total magnetic field yet they are referring to compressional pulsations. If the authors mean the Btot they should mention it along with the assumption that Btot is almost the same with Bcomp. Nevertheless, since they are showing the MFA coordinates later in the text, I don't understand the usage of these two figures especially when they also contain the x,y,z coordinates which are not discussed at all.

Lines 208-211: Please rename X-Y-Z to Poloidal-Toroidal-Compressional.

lines 271-274: The oscillations are of course visible but the rest of the statements are not supported by this plot as the reader can understand nor the exact frequency of these waves neither the phase difference. Maybe a simple filtering would give prominence to these pulsations or even better a spectral analysis.

Line 279: What is P and B? Please define.

Figure 13: There is a completely different behavior between low and high energy PA distributions yet the authors do not discuss it at all. I think there is much more information in this figure which should be further discussed.

Line 289: Please rephrase.

Lines 310-311: I don't understand this sentence. What do the authors mean by "most prominent". In figure 9, the double frequency is very pronounced from ~19:54 until after 20:32.

Lines 350-353: I would like to see the filtered time series of pressure or its fourier transform. As Kepko et al., 2002 have shown, the time interval that the authors examine is the ideal one for pulsations originating in the solar wind pressure.

Line 369: Please rephrase.
* * *

---

## Author Comment (AC1) · 17 Aug 2020

Dear Referee #1, Thank you for your comments and remarks. Below is our reply.

Lines 103-114: Please merge this chapter with introduction.

We have followed your suggestion and included the chapter in the introduction.

Lines 136-138: If it is not that important (I guess it is not since it is not shown) it shouldn't be mentioned at all.

We now include the Bz plot in Figure 1 as Bz is an important parameter in determining whether pulsations are produced by internal or external processes.

[Figure]

Line 152: Define cone angle. Also I don't see it anywhere in the figure. Instead I see the three components of speed which are not discussed at all. Figures are already too many (16!!!). Since Kp and speed are not discussed at all remove them and merge figures 1 and 2 (or even better provide the total solar wind speed only).

We followed your advice. We defined the cone angle and included it and the total solar wind speed in Figure 1. We merged Figures 1 and 2. We discussed the solar wind speed in the section on testing generation mechanisms. Now there are 14 figures in the paper.

Lines 170-178: The authors discuss the time-lag between solar wind pressure and compressions at GEO. Is this really important for the conclusions of this work? Moreover, I'm left with the feeling that the use of GOES measurements, in general, do not provide any significant observational evidence in this work. If I'm wrong then I believe that it should be discussed more clearly but if I'm not the authors should consider not using it at all.

The GOES data provide important spatial context for understanding this event. In particular, they help to contrast some of the features of the Pc4 and Pc5 waves observed at all four spacecraft. GOES data also confirm the value of data at geosynchronous orbit for monitoring solar wind conditions and the importance of solar wind pressure enhancements for stimulating and even amplifying compressional waves in the dayside magnetosphere. However, because of the much more complete instrumentation available on the Van Allen Probes, the remainder of this paper will focus on the observations from the latter spacecraft.

Lines 187-189: The authors state "Prior to the arrival of the strong solar wind dynamic pressure variations, RBSP-A observed very weak compressional pulsations with Pc5 periods and amplitudes of 1-3 nT from 18:15 to 18:55 UT." This is not shown anywhere.

The weak compressional pulsations observed by RBSP-A are not visible in Figure 3 because of the very large scale in this figure but readers can see some traces of them

in Figure 7.

In figures 5 and 6 the authors show the magnetic field in GSM coordinates along with the total magnetic field yet they are referring to compressional pulsations. If the authors mean the Btot they should mention it along with the assumption that Btot is almost the same with Bcomp. Nevertheless, since they are showing the MFA coordinates later in the text, I don't understand the usage of these two figures especially when they also contain the x,y,z coordinates which are not discussed at all.

We changed the text and Figures 4 and 5. Figure 5 shows only data in mean field-aligned coordinates, not including the field magnitude, and only for a shorter time interval. We discuss the pulsations in FAC and comment on the phase relations between the 3 components.

Lines 208-211: Please rename X-Y-Z to Poloidal-Toroidal-Compressional.

The definitions of the field-aligned coordinates that we used are precise, and are strictly based on observations. In the real rather than ideal magnetosphere, ULF waves, whether toroidal, poloidal, or compressional, often have at least some power in 2 or more components in a field-aligned system. Thus we do not label the three components of the magnetic field toroidal, poloidal, and compressional. Nevertheless, Pc5 pulsations are called compressional because of the prominent Bz component.

lines 271-274: The oscillations are of course visible but the rest of the statements are not supported by this plot as the reader can understand nor the exact frequency of these waves neither the phase difference. Maybe a simple filtering would give prominence to these pulsations or even better a spectral analysis.

We changed Figure 10 (formerly Figure 12) to show that that the intensities of electrons with energies from tens of keV to 2 MeV oscillate with Pc5 periods corresponding to those of the magnetic field. The energetic electron fluxes oscillated out of phase with the compressional Bz component of Pc5 magnetic field pulsations and did not display

any phase differences across all energies (see an expanded view for some selected energies, Figure 10b). We calculated the dynamic spectra for electrons at all available energies observed by RBSP-A and -B and found pulsations with frequencies similar to those for the compressional Pc5 pulsations (see Figure 10 A of this reply below). The lower energy electron fluxes displayed more noticeable enhancements as a response to the compressions of the magnetosphere.

Line 279: What is P and B? Please define.

We corrected the sentence: ...the antiphase relation between the plasma and magnetic field pressures suggests that particle pitch angle distributions peak near 90°.

Figure 13: There is a completely different behavior between low and high energy PA distributions yet the authors do not discuss it at all. I think there is much more information in this figure which should be further discussed.

We are not sure what feature in the figure the referee is addressing. We wish only to note: (1) the fact that pitch angle distributions peak near 90° pitch angles, (2) there are successive enhancements in response to compressions of the magnetosphere (for example at 1940 and 2005 UT at RBSP-A), and (3) these enhancements are most pronounced at the lower energies. At higher energies, flux variations associated with the radial gradients dominate the instrument response, as indeed can also be seen in Figure 10a.

Line 289: Please rephrase.

We slightly rephrased the sentence: The compressional components oscillated with a frequency twice that of the transverse component.

Lines 310-311: I don't understand this sentence. What do the authors mean by "most prominent". In figure 9, the double frequency is very pronounced from âĹij19:54 until after 20:32.

The referee's reading of Figure 9 (now Figure 7) is correct, but this sentence refers

to Figure 12, which presents the same data in a different format that does not include the relatively broad time window feature that is intrinsic to dynamic Fourier spectra. To prevent future confusion, we have added the following words at the end of this sentence: "in these line plots."

Lines 350-353: I would like to see the filtered time series of pressure or its fourier transform. As Kepko et al., 2002 have shown, the time interval that the authors examine is the ideal one for pulsations originating in the solar wind pressure.

We stated: First, with the exception of the interval from 19:35 UT to 19:55 UT, the Wind observations shown in Figure 1 provide no evidence for periodic solar wind drivers in the Pc5 range, be they density variations or IMF fluctuations, thus ruling out solar wind pressure pulses as the direct cause of the Pc4-5 pulsations. In what follows we show WIND data time-shifted 53 minutes (consistent with Figure 3), and confirm that solar wind pressure oscillations are not the direct cause of the Pc4-5 pulsations. Figure B of this reply compares dynamic spectra of the WIND solar wind pressure (shifted by 53 min) and of the RBSP-A total magnetic field from 17:30 UT to 22:00 UT on January 1, 2016. There is no evidence for significant solar wind pressure pulsations during this 4.5 hour interval. Only three very weak intensifications of pressure pulsation activity at Wind were observed during short intervals but they began 1.5 hour later than the generation of the magnetic Pc5 pulsations.

Figure C presents Wind time-shifted filtered data in the band of frequencies from 2 to 10 mHz. A monochromatic wave packet with frequency of ∼5 mHz only appeared between ∼20:30 and ∼20:50 UT that we marked in the Wind observations (not time-shifted) presented in Figure 1. We thus rule out solar wind pressure pulses as the direct cause of the Pc4-5 pulsations.

Line 369: Please rephrase.

We rephrased this sentence. As Figures 10 (a, b) demonstrate, RBSP-A shows no evidence in the electron observations for any such phase reversal at any relevant energy.

Thank you again for your help, Regards, Galina Korotova.

[Figure]

Figure A. Dynamic spectra for electrons with selected energies observed by RBSP-A from
18:30 to 21:00 UT and by -B from 20:40 to 23:00 UT on January 1, 2016.

[Figure]

**Fig. 1.**

[Figure]

Figure B. Dynamic spectra of the RBSP-A total magnetic field strength and the WIND solar wind pressure from 17:30 UT to 22:00 UT on January 1, 2016.

**Fig. 2.**

WIND    Time-Shifted Data    Jan 1, 2016    Yearday = 16001
Band Pass Filtered   f = 2 - 10 mHz   T = 1- 500 s

**Figure C**. Wind time-shifted filtered data in the band of frequencies from 2 to 10 mHz from 17:00 UT to 22:00 UT on January 1, 2016.

**Fig. 3.**

---

## Author Comment (AC3) · 17 Aug 2020

We included three new figures from the paper and some description of figure 4.

Figure 4a shows G-13 and -15 observations of the total magnetic field strength from 18:00 UT to 24:00 UT. The spacecraft observed long-duration Pc5 pulsations over a wide longitudinal region in the pre- and post-noon magnetosphere from 10:00 to 15:20 MLT (Figure 2). G-15 observed weak, less than ∼5 nT amplitude, Pc5 waves from 18:28 UT to 19:04 UT prior to the main event. During the main event from 19:04 to 23:00 UT, the magnetosphere was compressed (Figure 3), magnetic field strengths increased and the amplitude of these waves increased to values ranging from 10 to 16

[Figure]

nT with peak amplitudes prior to local noon. G-13 observed weak Pc5 pulsations with amplitudes of 2-4 nT throughout most of the time interval from 16:40 UT (not shown) to 21:00 UT. During the interval from 19:34 UT (∼14:45 MLT) to 20:10 UT (∼15:20 MLT), the pulsations reached slightly stronger amplitudes of 5-8 nT. At 23:02 UT all Pc5 wave activity at both GOES stopped. Figure 4b shows the RBSP-A and -B total magnetic field strength from 18:40 UT to 21:10 UT and from 20:40 UT to 23:10 UT, respectively, on January 1, 2016. Taken together, RBSP-A and -B observed Pc5 pulsations that occupied the inner dayside magnetosphere from 5.26 to 5.75 RE and from 09:56 to 12:44 MLT (Figure 2). Prior to the arrival of the strong solar wind dynamic pressure variations from 18:15 to 18:55 UT RBSP-A observed very weak pulsations with Pc5 periods and amplitudes of 1-3 nT (not visible at this scale). After the compression of the magnetosphere just after 19:00 UT, the pulsation amplitude at RBSP-A increased to values ranging from 10 to 15 nT with the peak amplitude occurring prior to local noon (Figure 4b). RBSP-B observed similar compressional Pc5 pulsations from 20:46 UT that ceased simultaneously with the end of the magnetospheric compression at about 23:02 UT.

[Figure]

Figure 1. Bz component of the magnetic field observed at Wind, and geomagnetic activity Dst and AE indices obtained from the OMNI database (upper panels) from 12:00 UT on December 30 to 00:00 UT January 2, 2016. The bottom panels show Wind observations of the magnetic field components, total magnetic field strength, cone angle, pressure, plasma density, and velocity from 16:00 UT on January 1, 2016 to 00:00 UT on January 2. 2016.

[Figure]

Figures 4 (a, b). G-15 and G-13 (a)  total magnetic field  strength  from 18:00 UT to 24:00 UT on January 1, 2016. RBSP-A  and -B (b)  total magnetic field strength  from 18:40 UT to 21:10 UT and from 20:40 UT to 23:10 UT on January 1, 2016, respectively,   Beneath the panels are listed the universal time (UT) and  magnetic local time  (MLT).

[Figure]

Figures 10 (a, b). RBSP-A observations of electron fluxes (a) in the energy range from 31.5 keV to 1704 keV from 18:30 UT to 21:00 UT and (b) their expanded view for selected energies from 19:20 UT to 20:00 UT.

---

## Author Comment (AC4) · 18 Aug 2020

Dear Referee,

We can not submit a revised manuscript now and think that information on slightly changed Figure 5 may be useful.

To determine the type of the Pc5 waves we converted the magnetic field observations from GSE into field-aligned coordinates (FAC). Here the Z axis lies parallel to the locally-averaged magnetic field. The Y axis points approximately azimuthally eastward and is transverse to B and to the outward radius vector. The X axis completes the

right-handed system and is directed approximately radially outward from Earth. Figure 5 presents RBSP-A and -B magnetic field observations in FAC. The Bz component is the value of the total magnetic field after subtraction of a 16-minute sliding average. The Pc5 pulsations are observed in all three components but the amplitudes of the azimuthal By and radial Bx components are rather small and do not exceed 7 nT. The compressional Bz component is much more pronounced for both spacecraft, reaching amplitudes of 14-15 nT before local noon, consequently, the pulsations are primarily compressional. The Bz component oscillated out of phase with the Bx component at RBSP-A and in phase at RBSP-B and in quadrature with the By component.

[Figure]

[Figure]

Figure 5. RBSP-A and -B magnetic field observations in field-aligned coordinates from 18:40
UT to 21:10 UT and from 20:40 UT to 23:10 UT on January 1, 2016, respectively.

---

## Referee Comment (RC2) · Anonymous Referee #2 · 14 Oct 2020

This paper uses field and particle measurements from GOES and Van Allen Probes to conduct a detailed exploration of dayside compressional waves observed during the geomagnetic storm of January 1, 2016.

This work examines latitudinal wave structure in terms of in-situ measurements, and demonstrate meridional oscillations ("sloshing") of the equatorial node about the equator in terms of frequency doubling observed in both the fields and particle signatures. The authors examine possible generation mechanisms for the compressional waves, and provide evidence that a mirror-mode instability is responsible for their generation.

Overall, I find the paper to be clear and well-written, and suggest only a few minor

revisions for publication.

- Line 58: "They have several Re wavelengths". Suggest "The have wavelengths of several $R_E$"

- Line 108: please define what you mean by "mode of the waves" and "nodal structure". Are we referring to azimuthal mode structure and latitudinal node structure? Or does "mode" refer to, e.g., compressional vs transverse waves?

- Line 138: since solar wind observations have not yet been introduced as a figure, suggest removing the words "(not shown)".

- Lines 163 et seq, and Figure 4. Please describe how the solar wind values are lagged. Is this a simple ballistic propagation estimation, a best fit estimation, or are propagation techniques such as those used in producing OMNI solar wind data used?

- Line 234: "min" -> "minute"

- Line 282: remove spurious period (".") between words "distribution" and "peak".

- Line 359: "Therefore, we conclude like many previous researchers that the...". Please provide citations for previous conclusions, or remove words "like many previous researchers".

- Lines 379 et seq. HOPE, EMFISIS, and RBSPICE contributions should be noted and described in Section 2, "Resources".

---

## Author Comment (AC5) · 18 Oct 2020

Dear Referee 2, Thank you very much for your comments and corrections. We have adopted all them. Enclosed, please find our replies to your remarks.

Line 58: "They have several Re wavelengths". Suggest "The have wavelengths of several $R_E$" – We corrected the sentence: They have wavelengths of several Earth radii.

Line 108: please define what you mean by "mode of the waves" and "nodal structure". Are we referring to azimuthal mode structure and latitudinal node structure? Or does

"mode" refer to, e.g., compressional vs transverse waves? – We changed the sentence as follows: We investigate the type of pulsation (compressional versus transverse), their harmonic mode, and their latitudinal nodal structure.

Line 138: since solar wind observations have not yet been introduced as a figure, suggest removing the words "(not shown)". We now show the solar wind observations in the final version of the paper.

Lines 163 et seq, and Figure 4. Please describe how the solar wind values are lagged. Is this a simple ballistic propagation estimation, a best fit estimation, or are propagation techniques such as those used in producing OMNI solar wind data used? –

To determine the lag time between the Wind and GOES-15 observations we related individual magnetosphere compressions to corresponding dynamic pressure variations. The good correspondence of GOES magnetic field enhancements to solar wind dynamic pressure pulses at the beginning and the end of the interval facilitated this task. Additionally, we confirmed these empirically derived lag times with simple ballistic estimates based on the solar wind velocity and the distance of Wind from Earth. Finally, we confirmed our estimates by examining the OMNI parameters.

Line 234: "min" -> "minute" – We changed min to minute

Line 282: remove spurious period (".") between words "distribution" and "peak". We removed period between words distribution and peak. The figure confirms that pitch angle distributions

Line 359: "Therefore, we conclude like many previous researchers that the...". Please provide citations for previous conclusions, or remove words "like many previous researchers". – We changed the sentence: Therefore, we conclude that the compressional Pc5 pulsations were excited by processes internal to the magnetosphere.

Lines 379 et seq. HOPE, EMFISIS, and RBSPICE contributions should be noted and described in Section 2, "Resources". We added additional descriptions of the RBSP

instruments. This paper employs observations of the most abundant ion components as well as electrons, over the 0.001–50 keV energy range of the core plasma populations from the HOPE instrument, populations of 20-4000 keV ion and electrons from the MagEIS instrument [Blake et al., 2013] in the Energetic Particle, Composition, and Thermal (ECT) suite [Spence et al., 2013], fluxes of ions over the energy range from âĹij20 keV to âĹij1 MeV and electrons over the energy range âĹij25 keV to âĹij1 MeV (RBSPICE) [Mitchell et al., 2013] in conjunction with observations from the magnetometer in the Electric and Magnetic Field Instrument Suite and Integrated Science suite (EMFISIS) [Kletzing et al., 2013], and the Electric Field and Waves (EFW) [Wygant et al., 2013] instrument. We examine electric and magnetic field measurements with 11 s and 4 s time resolution, respectively, and differential particle flux observations with ∼11 s (spin period) time resolution.